# Genomic Sub-Classification of Ovarian Clear Cell Carcinoma Revealed by Distinct Mutational Signatures

**DOI:** 10.3390/cancers13205242

**Published:** 2021-10-19

**Authors:** Douglas V. N. P. Oliveira, Tine H. Schnack, Tim S. Poulsen, Anne P. Christiansen, Claus K. Høgdall, Estrid V. Høgdall

**Affiliations:** 1Molecular Unit, Department of Pathology, Herlev Hospital, University of Copenhagen, DK-2730 Herlev, Denmark; douglas.nogueira.perez.de.oliveira@regionh.dk (D.V.N.P.O.); tim.svenstrup.poulsen@regionh.dk (T.S.P.); 2Department of Gynecology, Juliane Marie Centre, Rigshospitalet, University of Copenhagen, DK-2100 Copenhagen, Denmark; tine.henrichsen.schnack@regionh.dk (T.H.S.); claus.hogdall@regionh.dk (C.K.H.); 3Department of Gynecology, Odense University Hospital, DK-5000 Odense, Denmark; 4Department of Pathology, Rigshospitalet, University of Copenhagen, DK-2100 Copenhagen, Denmark; anne.pernille.christiansen.01@regionh.dk

**Keywords:** ovarian cancer, clear cell carcinoma, patient stratification, mutational signature, NGS, genomic instability, translational medicine

## Abstract

**Simple Summary:**

It is well-known that ovarian clear cell carcinoma presents unique molecular traits in comparison to other ovarian cancer subtypes, partially reflecting on chemotherapy resistance. Here, by investigating a panel of cancer-associated genes in a Danish population, our data showed that patients were further stratified into four different molecular subgroups: “PIK3CA”, “ARID1A”, “PIK3CA-ARID1A” and “Undetermined”, in regard to the presence of mutation on these genes. Somatic signature investigation further revealed that those subgroups share common features, such as ageing and defective MMR, whilst also bearing unique signatures. These findings suggest that different groups possess specific molecular backgrounds, indicating that such individuals could better benefit from more individualized therapy regimens.

**Abstract:**

Ovarian clear cell carcinoma (OCCC) is characterized by dismal prognosis, partially due to its low sensitivity to standard chemotherapy regimen. It is also well-known for presenting unique molecular features in comparison to other epithelial ovarian cancer subtypes. Here, we aim to identify potential subgroups of patients in order to (1) determine their molecular features and (2) characterize their mutational signature. Furthermore, we sought to perform the investigation based on a potentially clinically relevant setting. To that end, we assessed the mutational profile and genomic instability of 55 patients extracted from the Gynecologic Cancer Database (DGCD) by using a panel comprised of 409 cancer-associated genes and a microsatellite assay, respectively; both are currently used in our routine environment. In accordance with previous findings, *ARID1A* and *PIK3CA* were the most prevalent mutations, present in 49.1% and 41.8%, respectively. From those, the co-occurrence of *ARID1A* and *PIK3CA* mutations was observed in 36.1% of subjects, indicating that this association might be a common feature of OCCC. The microsatellite instability frequency was low across samples. An unbiased assessment of signatures identified the presence of three subgroups, where “PIK3CA” and “Double hit” (with *ARID1A* and *PIK3CA* double mutation) subgroups exhibited unique signatures, whilst “ARID1A” and “Undetermined” (no mutations on *ARID1A* nor *PIK3CA*) subgroups showed similar profiles. Those differences were further indicated by COSMIC signatures. Taken together, the current findings suggest that OCCC presents distinct mutational landscapes within its group, which may indicate different therapeutic approaches according to its subgroup. Although encouraging, it is noteworthy that the current results are limited by sample size, and further investigation on a larger group would be crucial to better elucidate them.

## 1. Introduction

Ovarian cancer is the eighth most common female malignancy, with approximately 295,000 new cases annually and 185,000 associated deaths [1,2]. Among its variants, epithelial ovarian cancer (EOC) is the most frequent histological subtype, of which ovarian clear cell carcinoma (OCCC) is characterized by a low sensitivity to chemotherapy; hence, it has the worst prognosis and a high frequency of associated deaths in late stage [3,4,5]. OCCC has varying geographical prevalence, ranging from 5–12% in Western countries to 10–30% in Asian countries, such as Japan and South Korea [6,7]. In Denmark, the Danish Gynecologic Cancer Database (DGCD) has reported a frequency varying from 2.1% to 3.4% in the last 5 years [7]. This database is a well-established compulsory national databank, which contains 97% of Danish patients diagnosed with gynecological cancer since 2005 [7,8]. The etiology is still elusive, although women with endometriosis have a higher risk of developing the disease [9]. Patients are diagnosed typically at a younger age (49–58 years old) and at early stage compared to other EOC subtypes [4,5,6,10], where stage I accounts for approximately 60% of all cases [9,10]. Hence, the 5-year overall survival in this subtype can reach up to 90% when found early [4]. Nonetheless, in those events where the disease is detected at a later stage, OCCC has the poorest prognosis among all EOC subtypes.

Next-generation sequencing (NGS) has revolutionized our understanding of cancer. It has enabled further classifications of tumor types defined by molecular features rather than only tissue of origin. These tools are now emerging as powerful means to better diagnose and treat cancers and are becoming routine clinical practice in oncology and pathology [11,12]. Furthermore, while whole exome/genome sequencing remains largely cost prohibitive [13,14], targeted gene panels have proliferated in mainstream clinical care due to their relative affordability and focused application, allowing for simplified data interpretation by omitting less relevant genes, reducing the number of variants of unknown significance, and increasing the depth of coverage to improve detection sensitivity [15]. In OCCC, there are limited investigations on its mutational landscape. Furthermore, most studies have been performed on global whole genome (WGS) or exome sequencing (WES) [16,17], which pose a big burden for clinical application due to the cost and time required for their data analysis. In the present study, we opted to use a panel of cancer-associated genes which are already employed in our clinical setup to examine a potential genomic sub-classification of OCCC by distinct mutational signatures. We further discuss the possible future clinical implications of further stratifying OCCC patients regarding therapeutic options, as well as the consequences of employing routinely used NGS panels for such classification.

## 2. Materials and Methods

### 2.1. Patient Cohort and Biological Samples

Patients diagnosed with OCCC and treated in the capital region of Denmark during 2005–2016 were identified in the DGCD. In the present study, we collected samples from two regional hospitals, Rigshospitalet and Herlev hospital. In total, 55 OCCC patients with corresponding data and tissue samples were included.

Biological samples were collected from the Department of Pathology, from Copenhagen University Hospital (Rigshospitalet, Copenhagen, Denmark) or Herlev University Hospital (Herlev, Denmark). Samples were prepared in formalin-fixed paraffin-embedded (FFPE) blocks at the time of primary surgery. In order to confirm tumor histology all preparations were reviewed by a specialized pathologist in gynecology. Histological diagnosis was classified following the latest World Health Organization classification. Tumor staging was performed according to the International Federation of Gynaecologists and Obstetrics (FIGO) guidelines.

### 2.2. DNA Extraction

Following identification of the tumor area, tissue samples were extracted by 1 mm disposable punchers. Genomic DNA was extracted by using Maxwell RSC DNA FFPE kit (Promega, Madison, WI, USA), automatized in Maxwell RSC (AS1450) instrument, following manufacturer’s instructions. Briefly, FFPE samples were pre-treated with mineral oil, followed by paraffin melting for 2 min at 80 °C and subsequent treatment with proteinase K and lysis buffer. DNA concentration was measured using the Qubit dsDNA HS Assay Kit (ThermoFischer Scientific, Waltham, MA, USA).

### 2.3. Microsatellite Instability Assay

Microsatellite analysis was performed by Plentiplex MSI kit (Pentabase Aps, Odense, Denmark), following manufacturer’s instructions. The MSI panel primers is comprised of 5 primer sets (BAT25, BAT26, D2S123, D5S346, and D17S250). Microsatellite loci were individually evaluated by comparing the length of amplicons from reference DNA to those from the samples, using both shorter and longer microsatellite amplicons as indicative of instability. Patients were characterized as microsatellite-stable, MSS (no presentation instability) or microsatellite instability-high, MSI-H (3 or more unstable loci).

### 2.4. DNA Next-Generation Sequencing

We employed the Oncomine Tumor Mutation Load (TML) Assay (ThermoFischer Scientific, USA), a panel comprised of 409 oncogenes relevant across major cancer types, including tumor mutational load assay. Then, 100 ng of DNA for each sample were amplified using the Ion AmpliSeq library preparation kit 2.0 (ThermoFisher Scientific, USA), followed by Ion Library Equalizer kit in order to assure equal chip loading. Sequencing was performed on the IonTorrent S5 XL platform, following manufacturer’s instructions. Sequencing data were acquired, pre-processed, aligned and analyzed by Ion Suite Software v5.6 (ThermoFisher Scientific, USA). Further filtering for true positive variants was processed using R environment, described below. Finally, all called variants were manually reviewed and visually inspected by integrated genomic viewer (IGV) (Broad Institute, Cambridge, MA, USA). Only variants in exonic regions or splice variants were selected.

### 2.5. Data Analysis

Primary NGS data analysis was performed by using the Ion Reporter (ThermoFisher Scientific) TML algorithm version 2.5. Briefly, read ends from raw FASTQ files were trimmed to avoid poor quality sequencing or adapter contamination. Sequences were aligned using hg19 genome build as reference. BAM files were then submitted to the Ion Reporter for variant calling, with a threshold for minor allele frequency (MAF) of 5%. Variants were further submitted to a second technical filtering round comprised of removal of variants with depth below 100 reads and below 10% of the median of variants from the same sample, Phred score <100 and Phred strand bias score >60. Furthermore, due to an inherited feature of the IonTorrent platform, regions containing at least 6 homopolymers were also removed. During annotation, variants identified as potential germlines (based on 1000 Genomes and GnomAD consortia), common SNPs (based on dbSNP and COSMIC databases) were also excluded. Tumor mutational load (TML) was determined by the Mutation Load filter module, as claimed by the manufacturer. The calculation was performed as follows:$$\mathrm{TML}=\frac{\mathrm{Somatic}\;\mathrm{mutation}\times 10^{6}}{\mathrm{Total}\;\mathrm{exonic}\;\mathrm{bases}\;\mathrm{with}\;\mathrm{sufficient}\;\mathrm{cover}}$$

In accordance with our current clinical cut-off, samples with TML equal or above 20 were classified as “high TML”, whilst the reminder as “low TML”. The second filtering steps were performed in R environment [18].

For somatic signature analysis, we used non-negative matrix factorization (NMF) method in order to infer the number of signatures within our cohort, irrespective of any clinical information, as previously described [19,20]. Briefly, the method consists of evaluating the base variant in the context of its immediate neighboring bases. For instance, if a transition mutation of C to T occurs in the sequence GCA, that mutational motif is defined as G[C>T]A. Thus, the frequencies of the 96 (4 × 4 × 4) possible motifs across all samples define the mutational spectrum, which can be associated with different cancer types or disease etiologies. Considering the four main subgroups found, cophenetic correlation was applied to define the number of potential mutation signatures, ranging from 1 to 4. NMF was repeated 500 times for each value of possible signatures. Results indicated a lack of robustness when a decomposition exceeded three signatures in this cohort. As such, we selected to decompose the pooled mutation data set into three stable mutational signatures. For Kataegis analysis, all mutations from the cohort were ordered by chromosomal position and the intermutation distances were calculated as the number of base pairs (bp) from one mutation to its next. Those distances were further segmented using constant fitting to find regions of constant intermutation distances. Putative regions of kataegis were identified as those segments containing six or more consecutive mutations with an average intermutation distance of ≤1000 bp [21].

### 2.6. Statistical Analysis

The calculation of strand bias applied on the NGS data filtering was performed by Fisher’s Exact Test and converted into Phred scale (PS), as described below:$$\mathrm{PS}=10\times \log_{10}p.\mathrm{value}$$

APOBEC enrichment score was estimated by the following formula, as described previously [22]:$$E=\frac{\left({\mathrm{mut}}_{\mathrm{tCw}}\times {\;\mathrm{background}}_{c\left(\mathrm{or}\;g\right)}\right)}{\left({\mathrm{mut}}_{C\left(\mathrm{or}\;G\right)}\times {\;\mathrm{background}}_{\mathrm{tcw}}\right)}$$
where: mut_tCw_—*n* of mutated cytosines (and guanines) flanked by tCw (or wGa) motif; mut_C(or G)_—total *n* of mutated cytosines (or guanines); background_c(or g)_—total *n* of cytosines (or guanines) flanking 20 bases around mutated cytosines (or guanines); background_tcw_—total *n* of tcw (or wga) flanking 20 bases around mutated cytosines (or guanines)

Samples with an enrichment score above 2 were classified as APOBEC-positive. For somatic signature assessment, data was analyzed by NMF decomposition, Residual Sum of Squares (RSS) and explained variance statistics. All statistical analyses were performed in R environment [18], with implementation of SomaticSignatures [23], YAPSA [24] and KataegisPortal [21] packages.

## 3. Results

### 3.1. Molecular Stratification and Localized Hypermutation

We performed a panel-based NGS, comprised of 409 cancer-associated genes in tissue samples from 55 OCCC patients in order to assess their molecular profiling, with an average coverage of 732× (210–1740×). In total, 982 variants were identified, where 267 mutations were identified as silent, whilst 615 were missense, 29 nonsense, 29 frameshift, 21 in-frame, 17 splice-site or -region, 3 intronic and 1 translation start site (TSS). Among the substitution types, transitions and transversions represented an average of 70.3% and 29.7%, respectively. In total, 98.2% of the tissue samples (54/55) presented with at least one single nucleotide variant (SNV). The complete spectrum of mutation variants and types is depicted in Figure 1A. In line with previous reports, *ARID1A* and *PIK3CA* were the most prevalent mutations observed, present in 49.1% (27/55) and 41.8% (23/55), respectively (Figure 1B). From those, *ARID1A* and *PIK3CA* mutations co-occurred in 36.1% (13/36) of the samples. The mutation frequencies were followed by *SYNE1* (20%), *KMT2C* (18.2%), *KMT2D* (18.2%), *CSMD3* (18.2%), *PDE4DIP* (14.6%), *ATM* (12.7%), and *CYP2D6* (12.7%). Approximately 44.4% of all mutations found on *ARID1A* were characterized by a deletion of glutamine on position 1334 (*ARID1A*^Q1334del^), and in *PIK3CA,* 47.8% by a substitution from histidine to proline at position 1047 (*PIK3CA*^H1047P^). In all other genes the mutations were sporadically distributed. Moreover, 14 samples (25%) were identified as high TML, and 4 (7.1%) with MSI-H, indicating high genomic instability.

Considering that *foci* of localized hypermutation events, known as kataegis, have been previously reported in cancer cases, including OCCC, we assessed whether they also manifested in this cohort. Fourteen kataegis *foci* were identified in chromosome three, seven on chromosome 7, and seven on chromosome 22. All events on chromosomes 7 and 22 were detected in the promoter region of *KMT2C* and *CYP2D6*, respectively (Figure 1C). Given that the association of APOBEC and kataegis events has been previously suggested, we estimated the APOBEC enrichment score across all samples [22]. In total, 16.7% of the samples presented with an APOBEC enrichment. We also observed that the frequency of mutations was higher in this group, compared to the non-enriched group (Figure 1D). Interestingly, APOBEC enrichment and high TML did not co-occur in this cohort (*p* = 0.21). On the other hand, 75% of MSI-H cases did also present with high TML, indicating a concomitant onset (*p* < 0.044) (Figure 1B).

### 3.2. Co-Occurrence of Mutations and Metabolic Pathways

Given the complexity of cancer network, the mutation pattern in certain genes tend to co-occur with other genes, suggesting that the association might contribute to the initiation and development of the disease. Similarly, genes can also occur in a mutually exclusive manner, indicating their similar role in a common pathway. Based on the most frequently mutated genes, we were able to identify 48 co-occurring events, such as seven with *KMT2D*-*PIK3CA*, four with *KMT2C*-*MYH9*, four with *KMT2C*-*ATM*, three with *MYH9*-*TAF1L*, among others (Figure 2A; Appendix A). On the other hand, 25 mutually exclusive events were identified, among three pairs of genes, 10 in *PIK3CA*-*SYNE1*, 7 in *PIK3CA*-*ATM*, and 8 in *ARID1A*-*PKHD1*. Next, functional pathway analysis was performed in order to understand whether the mutated genes were associated in common pathways. We identified three major pathways, namely TRK-RAS, PI3K and NOTCH, present in 64.3%, 62.5% and 35.7%, respectively (Figure 2B).

Furthermore, we identified potential clinically druggable targets based on the Drug Gene Interaction database, DGIdb [25] (Figure 2C). In total, we identified 9 clinically actionable targets, represented mainly by 4 categories: histone modification (*KMT2C*, *KMT2D*, *EP400*, *TAF1L* and *ATM*), kinase inhibitors (*PIK3CA*, *ATM*, *PRKDC*, *TAFL1* and *PPP2R1A*), DNA repair (*ATM*, *PRKDC*, *RECQL4* and *TRRAP*), tumor suppressor (*ATM*, *PRKDC*, *PKHD1* and *PPP2R1A*).

### 3.3. Mutational Signatures in OCCC

Considering the prevalence of mutations in the *ARID1A* and *PIK3CA* genes, we observed 4 seemingly distinct classifications in our cohort: patients with only *ARID1A* mutation (“ARID1A” subgroup), only *PIK3CA* mutation (“PIK3CA” subgroup), *ARID1A* and *PIK3CA* double mutation (“Double hit” subgroup) and those with no mutations on either genes (“Undetermined” subgroup) (Figure 1B). Mutational signatures may reflect specific underlying factors associated with tumor development. Therefore, in order to evaluate whether those subgroups were characterized by different mutational spectra, we performed mutational signature analysis. The most frequent mutations occurred on the N[C>T]N context (where N is any of the 4 bases) in all subgroups. Furthermore, ARID1A-, PIK3CA- and Double hit subgroups suggested enrichment on the N[T>C]N context, whilst the ARID1A- and Undetermined subgroups on N[C>A]N, but those differences were not confirmed by multiple comparison with adjusted *p*-values (Appendix A; Appendix A). Nonetheless, when submitted to an unbiased *de novo* assessment of signatures, our classification showed the presence of only three (explained variance >98%), with PIK3CA- and Double hit-subgroups exhibiting unique signatures, signature2 and signature3, respectively (Figure 3A,B). The ARID1A and Undetermined subgroups showed similar profiles, indicating that the OCCC histological subtype might be overall characterized by three different mutational landscapes.

In order to identify potential associated factors that might play a role in OCCC, we compared the four subgroups with the compendium of mutational signatures from the COSMIC database. As shown in Figure 3C, the ageing signatures, SBS1 and SBS5 from COSMIC, were the most evident across the four subgroups. A high prevalence of defective mismatch repair (MMR) signature, SBS6, was also observed in all groups, to a lesser extent in the Double-hit subgroup. MMR signature is associated with defective DNA mismatch repair and is commonly found in microsatellite unstable tumors. On one hand, our results showed low prevalence of MSI-H cases, irrespective of subgroup. On the other hand, in line with our findings, MMR signature has been observed with high prevalence in OCCC subtype, albeit no presence of microsatellite instability [26]. The presence of the APOBEC signature, SBS2 and SBS13, was higher in the Double-hit subgroup as compared to the others, constituting its second strongest feature. The PIK3CA subgroup was mainly represented by the defective MMR signature, and by a minor, but exclusive association with the reactive oxidative species (ROS) damage signature, SBS17. Similar to this subgroup, ARID1A and Undetermined subgroups also presented a high exposure to MMR signature, except that in both the APOBEC signature was also more prevalent than in the PIK3CA subgroup. Moreover, exposure to defective POLE mutational signature, SBS10, was also observed to a low degree in the Double hit and Undetermined categories. Here, the mutational spectrum further suggests the underlying molecular differences among these four subtypes, whilst indicating the uniqueness of Double hit and PIK3CA subgroups and similarities between ARID1Aand Undetermined subgroups.

## 4. Discussion

OCCC is well known for presenting unique molecular features in comparison to other EOC subtypes. In contrast to other subtypes, OCCC usually does not show frequent mutations on the *TP53* gene, and very low prevalence of *BRCA1* and *BRCA2* truncations [27,28]. Those differences are also reflected in the clinical outcome. Patients with OCCC are more likely to be diagnosed at an early stage, with good prognosis when confined in the ovary. However, when diagnosed in advanced stages the prognosis is poor, due to resistance to chemotherapy [9,27]. OCCC incidence is also particular to geographical prevalence. In Denmark, the incidence of OCCC frequency has been reported from 2.1% to 3.4% in the last 5 years [7]. Hence, the studies performed in this subtype is limited and heterogeneous. Most of the investigations in regard to OCCC genomic landscape were performed in mixed EOC cohorts, focused on subtype stratification [26,29]. Nonetheless, recent works have shown the genomic features unique to OCCC [16,17].

Whole genome- and exome-sequencing have emerged over the last decade and proven as invaluable tools for assessment and discovery of genomic features in a plethora of cancer types. These methods have been used to identify several potential driver mutations, as well as clinically actionable variants [13,15]. In OCCC, the few studies assessing genomic mutations were performed on those platforms [16,17,30]. Nonetheless, from a clinical and economic perspectives, the massive amount of data generated by WGS/WES is overwhelming. Indeed, more than 95% of the data are not of therapeutic relevance. Coupled with that, the time required for performing the sequencing and data analysis make wholesome genome assessment inefficient in a clinical setting [13]. Hence, the use of validated cancer-associated gene panels provides a cost- and time-effective NGS technology, with direct clinical use. Here, by using a panel comprised of 409 genes, we assessed the genomic characteristics of OCCC. As reported previously, we observed that *ARID1A* (49.1%) and *PIK3CA* (41.8%) were the most prevalent mutations [16,17,31,32]. Interestingly, given the population heterogeneity of the current and previous studies, the high presence of *ARID1A* and *PIK3CA* variants suggests that these are features specific to OCCC, rather than population ethnicity. *ARID1A* constitutes the main catalytic subunit of the chromatin remodeling complex SWI/SNF [33,34]. In line with our findings, early studies have shown that the vast majority of mutations (above 95%) in *ARID1A* were inactivating, mainly represented by nonsense or frameshift variant mutations [32,33]. Those mutation frequencies were accompanied by decreased expression of its translated protein [32,35], indicating a direct association between the mutational status and protein expression. A plausible explanation might be an epigenetic-driven silencing of ARID1A protein expression in the context of a heterozygous mutation, without necessarily the canonical loss of heterozygosity. In accordance with this hypothesis, by investigating a cohort of OCCC, Wiegand and collaborators found that 73% of those samples with *ARID1A* mutation also showed loss of its correspondent protein expression [36]. Further, their immunohistochemistry staining in a validation cohort showed concordant results. Taken together, it is possible that post-transcriptional mechanisms account for loss of ARID1A protein in OCCC harboring heterozygous mutations, based on the fact that RNA sequencing was able to detect both wild-type and mutant alleles in their cohort. Unfortunately, we were not able to examine RNA expression in our cohort in order to confirm such results. Interestingly, we found an abnormal high prevalence of *ARID1A*^Q1334del^, accounting for 46.2% of the cases. An in-frame mutation at that residue has been previously reported on OCCC, namely *ARID1A*^Q1334_R1335dupQ^, whilst the Q1334del variant have been shown in other types of cancer [37,38]. To date, its functional role and implication is still elusive. Nonetheless, in line with most of *ARID1A* mutations in OCCC, it is possible that *ARID1A*^Q1334del^ leads to an inactivating mutation, impairing protein expression. Moreover, alterations in the SWI/SNF complex have been found in high frequency in OCCC cases. In a WES study, Itamochi and colleagues reported that in 51% of the cases, at least one gene of that pathway was mutated. *ARID1A* had the highest prevalence, with 42%, followed by *ARID1B* (18%), and to a lesser extent by *SMARCA4* (5%) and other genes [16]. Noteworthy, except for *SMARCA4*, the gene panel used in our study does not contain all genes associated with SWI/SNF. Hence, we were not able to observe an enrichment of this pathway in our analysis. Nevertheless, the alteration frequency of *SMARCA4* (3.6%) in our cohort was similar to that reported previously.

Similarly, the frequency of *PIK3CA* mutations was also high in the current cohort. Mutations in this gene have been shown to be strongly associated with tumor development, through activation of the PI3K/mTOR pathway. In that regard, we showed that 62.5% of individuals presented at least one truncated gene from the PIK3 signaling pathway. By its constitutive activation, this complex favors uncontrolled cell proliferation. The frequency of *PIK3CA* (41.8%) mutations in the present cohort is in line with results from previous studies. Most mutations were characterized as *PIK3CA*^H1047P^, accounting for 56.5%. This variant is a well-known hotspot within the *PIK3CA* kinase domain, which leads to a catalytically active protein [39]. The co-occurrence with *ARID1A* (in 50% of the cases has also been described before and seems to be a specific molecular feature of OCCC. A pan-cancer analysis in data sets from The Cancer Genome Atlas (TCGA), revealed that the co-occurrence of *ARID1A*-*PIK3CA* was highest in OCCC cases, representing 33% of all cases as compared to other cancer types [32,40]. Interestingly, the co-existence of mutations on *ARID1A*-*PIK3CA* was shown to be associated with formation and development of OCCC in mice model. Chandler and colleagues were able to generate *ARID1A* conditional allele, where they observed that although loss of *ARID1A* promoted embryonic lethality, it did not lead to tumor formation by itself [40]. That indicates that *ARID1A* requires an additional mutation in a proto-oncogene in order to initiate tumorigenesis. Indeed, they further generated *PIK3CA*^H1047P^ variant in that mice model and observed that: (1) similar to *ARID1A*, *PIK3CA* mutation was not sufficient to initiate OCCC tumor formation, and (2) co-occurrence of both variants led to the rapid initiation of ovarian tumorigenesis, with peritoneal metastases observed in approximately 50% of the cases [40]. Taken together, it can be speculated that dysfunctional *ARID1A* and *PIK3CA* are likely oncogenic factors driving formation of OCCC. From a clinical perspective, that indicates that patients with *ARID1A* mutation could potentially benefit from PIK3 inhibitors. Moreover, experiments using transient depletion of *ARID1A* showed that those cells are significantly sensitive to PIK3/AKT kinase inhibitors [41]. There is currently one ongoing clinical trial investigating the use of kinase inhibitors in OCCC patients, where mutations on *ARID1A* and *PIK3CA* will be evaluated (NCT01914510). In a similar manner, patients harboring mutations on *PIK3CA* could potentially benefit from specific inhibitors. In that regard, recent results indicate that its inhibition efficiency might be specific to the presence of other co-occurrent mutations. The application of AZD8835, AZD5363 and AZD8186 in OC mouse model showed that the latter was not sufficient to stall tumor formation. Furthermore, although AZD8835 and AZD5363 were efficient in reducing tumor progression, the use of AZD8835 further inhibited the activation of *BRCA1/2* mRNA expression in *KRAS*-mutated cell lines [42]. Furthermore, our results also showed some common druggable targets for OCCC, such as histone modifications (e.g., *KMT2C*, *KMT2D*) and kinase activity (e.g., *ATM*).

We further investigated the mutational spectrum in order to identify potential unique patterns within the same disease. An unbiased *de novo* signature assessment identified 3 distinct subgroups in this cohort. We observed that the “Double hit-” and the “PIK3CA-” subgroups presented with a unique pattern of mutation landscape, whereas individuals from the “ARID1A-” or “Undetermined-” subgroups were very similar to each other. In regard to the latter observation, it might underscore the understanding that loss of *ARID1A* is important but not sufficient for the tumorigenesis process. On the other hand, the unique mutational pattern presented by the PIK3CA subgroup might indicate that malfunctioning of this gene is essential during the early manifestation of OCCC. However, it requires mutations on other passenger genes for the disease progression. In that regard, it has been shown that the loss of *PIK3CA* function led to multifocal sites of epithelial hyperplasia but with no development to tumor formation [40]. Moreover, we observed that the most prevalent pattern across all subgroups was associated with ageing, SBS1 and SBS5. This signature is commonly found in most types of cancer, and its prevalence is known to associate with fast cell division and proliferation. Noteworthy, the SBS1 signature is also associated with deamination of cytosine, a process that can be induced by FFPE preparation. Thus, we cannot dismiss that the high prevalence of the observed signature might be partially due to the sample storage method. In line with previous findings, the APOBEC signature was found in the present cohort, but with relevant differences across the subgroups [26]. Here, its presence was proportionally similar to that of defective MMR in the “Double hit” subgroup, whilst its presence was the lowest in the “PIK3CA” subjects, and similar in the remaining “ARID1A-” and “Undetermined” subgroups. Noteworthy, the signature for ROS damage, SBS17, was exclusive to “PIK3CA”. The association between the PIK3/AKT and ROS pathways might be the signaling or initiation of tumor development. An early study has shown that the activation of the former pathway lead to the formation of ROS through activation of its initial cascade, NADPH oxidase activation [43]. Others suggest that formation of ROS and activation of PIK3/AKT pathway are interdependent in a feedback loop, which in turn promotes survival of leukemic cells [44]. Except for the “Double hit” subgroup, we also observed that the featuring signature of defective MMR, SBS6, is strongly prevalent in OCCC. Deficiency on this repair system is a hallmark of increased formation of microsatellite frequencies in tumors, and it has been associated with OCCC previously [45,46]. Interestingly, mutations on any of those genes were not among the most prevalent, nor was a ubiquitous incidence of MSI-H detected in our cohort. We speculate that the activity of MMR might be controlled by epigenetic regulation. First, apart from the high presence of *ARID1A* mutations, we also observed an unusual high prevalence of variants on other epigenetic factors, such as *KMT2C*, *KMT2D* and *EP400*. Second, methylation on the promoter region of *MSH1* has been shown to disable MMR through transcription downregulation of its target [47,48]. In corroboration with our results, despite impairing MMR function, the methylation of MSH1 did not lead to an increased frequency of MSI [47]. Finally, considering that the chromatin remodeling system has been shown to be associated with OCCC, it will be interesting in the future to assess the genome-wide DNA methylation profile in such subjects.

Noteworthy, by using WES approach, Shibuya and collaborators identified subgroups of OCCC patients based on hierarchical clustering of base substitution frequencies [30]. They were able to identify a subgroup with higher APOBEC prevalence. In comparison to their study, we performed OCCC stratification primarily based on the presence (or absence) of *ARID1A* and *PIK3CA* alterations. Furthermore, the molecular signatures were identified on the context of the single variant and its flanking bases, as described previously [20]. The rationale for our approach was that those signatures are constantly curated, independently validated by other studies, and updated by the COSMIC database. From the clinical perspective, the identification of variants and signatures with known deleterious effect on the patient is important in order to improve relevant therapeutic regimen. Furthermore, we focused on the feasibility of identifying OCCC subgroups performed by a cancer-associated gene panel routinely used in our clinical setting, with the perspective of a seamless future adaptation of this method to the current workflow. Notwithstanding, we have not compared this approach from that presented by Shibuya and colleagues for performance evaluation. In that regard, apart from the time and cost discussed above, the use of WES provides a more comprehensive mutational landscape, with thousands of variants not covered by a gene panel, whilst the latter is more focused on targets with known druggable actions with a higher confidence provided by high read depth.

In summary, the current findings suggest that OCCC presents distinct mutational landscapes within its group, which may indicate different therapeutic approaches accordingly. For instance, the “Double hit-” and “PIK3CA” subgroups display their own distinctive signatures, which might indicate susceptibility to PI3K- or mTOR-inhibitors. In contrast, the “ARID1A” subgroup could respond better to HDAC inhibitors. In that regard, preliminary reports have shown that *ARID1A*-mutated OCCC patients were described as being highly sensitive to HDAC2 and HDAC6 inhibitors [49,50]. Furthermore, one of the strengths of this study is that we focused in using a panel of targets routinely employed in our clinical setting, in order to evaluate its potential application in the future. Hence, we were able to confirm what has been previously reported in OCCC molecular profiling [16,17,30,32], while indicating that the stratification of OCCC subgroups has the potential feasibility to be applied in a clinical setting, which could benefit patients with better individualized precision medicine. Although encouraging, there is limited resources on OCCC genomic investigations to date, with no public data available, which impaired the validation of our finding in independent cohorts. It is important to note that our results are also limited by a relatively small sample size, and further investigation on a larger group is required. Nonetheless, the current findings can hopefully contribute to further studies in the area for future targeted therapy.

## 5. Conclusions

Among epithelial ovarian cancer subtypes, OCCC presents with unique molecular features in comparison to other subtypes, such as a lack or low frequency of *TP53*, *BRCA1* and *BRCA2* mutations, of which mechanisms are hitherto elusive. Here, by investigating the mutational spectra we were able to identify distinct subgroups of OCCC. The results indicated common features among them, such as ageing and defective MMR signatures, as well unique representation, such as APOBEC enrichment in the “Double-hit” subgroup and ROS in the “PIK3CA”. In contrast to other subtypes, our findings also showed that MSI-H are not frequent in OCCC, which might contribute to resistance to standard chemotherapy regimen. Taken together, these findings suggest that the stratification of patients according to their mutational signatures may improve treatment outcome by specific therapy regimens. Nonetheless, the current results should be interpreted with discretion, given its cohort size.

## Figures and Tables

**Figure 1 cancers-13-05242-f001:**
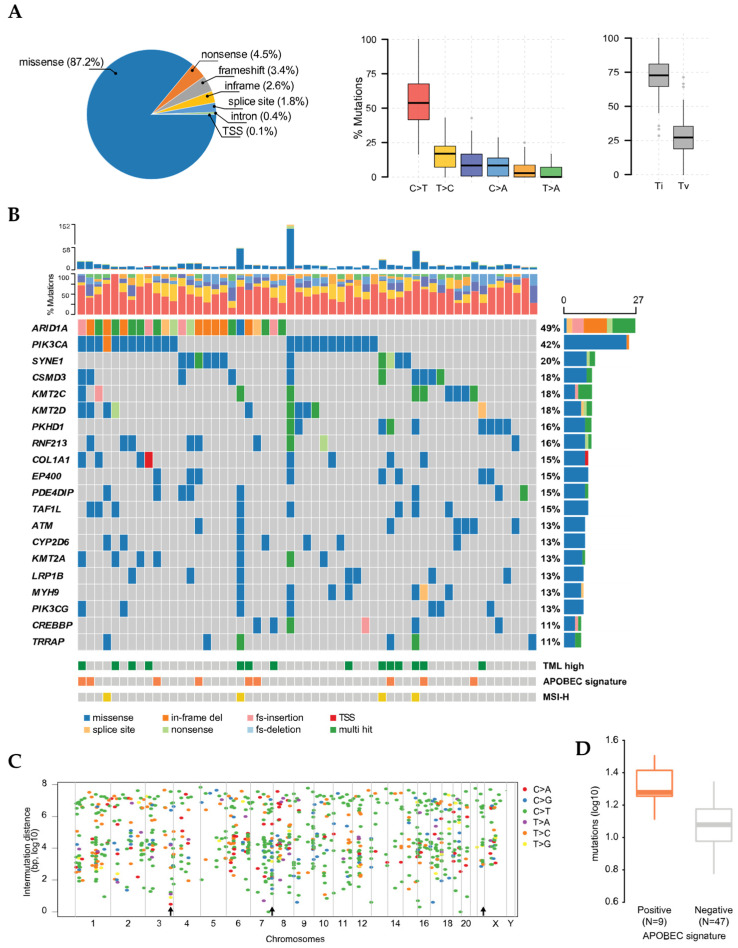
Overall mutational landscape in OCCC. (**A**) Types of mutations represented in the cohort, composition of base substitutions and classification (Ti: transition; Tv: transversion). (**B**) Oncoplot for each patient, with the most frequent mutated genes, TML status, APOPEC signature and MSI classification (TML: tumor mutational load; MSI-H: microsatellite instability high). On the top panel, the bars show the variant type for each individual, whereas the middle panel depicts the frequency of base substitution. (**C**) Intermutation distances, indicating kataegis events on chromosomes 3, 7 and 22 (black arrows). (**D**) Mutational load in APOBEC-positive (orange bar plot) and negative (grey bar plot) groups. TSS: transcription start site.

**Figure 2 cancers-13-05242-f002:**
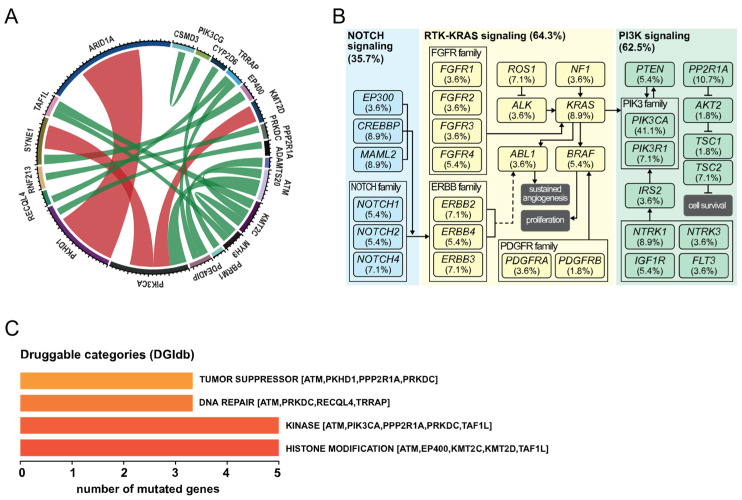
Mutation co-occurrence and pathways misregulation in OCCC. (**A**) Circos plot diagram depicting the pairwise co-occurrence of gene mutations (green arches) or mutual exclusivity (red arches). (**B**) Representation of the most affected pathways, NOTCH (35.7%; blue), RTK-RAS (64.3%; yellow) and PI3K (62.5%; green). Each box represents the mutated gene found in the cohort and its frequency. (**C**) Integrative analysis with DGIdb showing the potential druggable categories. Overall mutational landscape in OCCC.

**Figure 3 cancers-13-05242-f003:**
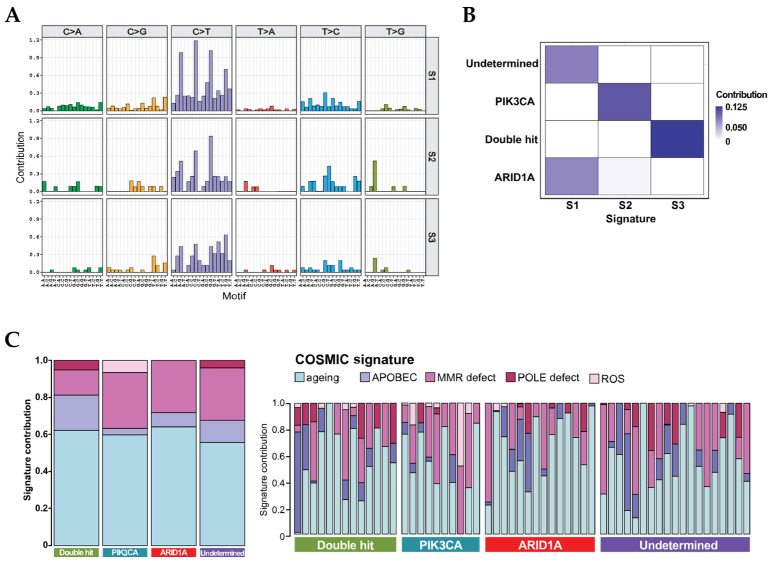
Patients show distinct mutational signatures within OCCC. (**A**) De novo assessment of mutational landscape indicated that OCCC patients presented 3 different signatures, signature1, signature2, and signature3. The sub-groups “Double hit” and ”PIK3CA” showed unique signatures, signature2 and signature3, respectively. Whereas “ARID1A” and “Undetermined” sub-groups shared the same signature, signature1 (**B**). (**C**) COSMIC supervised signatures within the 4 sub-groups, confirming the differences among them. The highest relevant of signatures were from ageing, APOBEC, MMR defect, POLE defect, and ROS. Mutation co-occurrence and pathways dysregulation in OCCC.

## Data Availability

Data available upon reasonable request from the authors, due to privacy/ethical restrictions.

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
