# Peer review of "Genomic Sub-Classification of Ovarian Clear Cell Carcinoma Revealed by Distinct Mutational Signatures"

_cancers, 2021, doi:10.3390/cancers13205242_

Round 1

Reviewer 1 Report

This paper seems to be based on the DGCD containing 97% Danish patients which need to be clearly mentioned in the title and/or abstract. 

Do you think that the mutations found in the study can be generalized to a demographically diverse population or are limited to the Danish population?

Can you provide the list of details of the patients or something similar to your previous paper? See attachment.

Author Response

Please find the enclosed file with the respective answers to the Reviewer.

Reviewer 2 Report

In this study, the authors investigated by NGS analysis the specific molecular features of ovarian clear cell carcinoma (OCCC), characterized by dismal prognosis, partially due to low sensitivity to standard chemotherapy regimen. In the series of OCCC patients analyzed by investigating a multigene panel of cancer-associated genes, they identified 4 different molecular subgroups: “PIK3CA”, “ARID1A”, “PIK3CA-ARID1A” and “Undetermined”, in regard to the presence of mutation on such genes. Authors conclude that this analysis significantly suggests specific mutational landscape within each group identified indicating that such individuals could better benefit from different therapy regimens.

Main points:

  • In the Methods section, please better describe the “Microsatellite instability assay” section, with particular regard to the number of total analyzed MSI loci. In addition, in the “Data analysis” section, please add the reference or methods developed and used to calculate Tumor Mutational Load (TML) values and to determine the optimal cut-off of TML values. In the “Statistical analysis” section, please insert the full terms for the abbreviation “RSS”, at the end of the line 169.
  • In the Results section (“Mutational signatures in OCCC”), please better describe the prevalence of defective mismatch repair signature among the molecular subgroups.
  • Probably, in the discussion section, the authors can add a description of the different therapy regimens suitable for specific molecular subgroups and alternative to chemotherapy.

Minor points:

  • In the abstract section, please insert the abbreviation for Ovarian Clear Cell Carcinoma. In addition, probably you can insert here also the results about the microsatellite instability analysis and a brief description of the molecular characteristics of the groups designed as “Double hit” and “Undetermined” (with and without mutations on ARID1A and PIK3CA genes, respectively).
  • In the discussion section, delete the bracket at the end of line 352.

Author Response

(The authors gave the same response as above.)

Reviewer 3 Report

Oliveira et al. analyzed DNA next generation sequencing (NGS) and microsatellite instability assay from 55 patients of ovarian clear cell carcinoma (OCCC). Although the manuscript was well described, there are some major points to be revised.

Major points

  1. Materials and Methods

2.1. Patient cohort and biological samples

1) The 55 patients included in this study. Although the OCCC is relatively a rare ovarian cancer subtype in western countries, however, this sample size is relatively small for the period of study (2005-2016; Gaducci et al. Gynecol Oncol. 2021 Sep;162(3):741-750.).

  1. Results

3.1. Molecular stratification and localized hypermutation

1) In figure 1A, the authors showed types of mutations represented in this cohort. However, mutation types/variants in individual patients are not described. Please consider showing mutation types and variants in each patient (van Riet et al. Nat Commun. 2021 Jul 29;12(1):4612.).

2) The clinical background data is not shown in this manuscript. Considering clinical application of this distinct mutational signatures, the patients background data should be presented in a figure, i.e. overall survival, FICO staging, age, race and endometriosis (Khalique et al. Cancers. 2021 Jul 30;13(15):3854.).

3.3 Mutational signatures in OCCC

1) The authors showed ARID1A-, PIK3CA and double hit subgroups showed enrichment on the N[T>C]N context whilst the ARID1A- and Undetermined subgroups were enriched on N[C>A]N. The readers may want to know whether these enrichments were significant difference among the subgroups. Please consider performing statistical analysis for analyzing frequency of these enrichment among subgroups.

2) In this cohort, there is no survival and treatment data available. ARID1A, PIK3CA mutation seems correlated to shorter survival and chemo resistance. Please show the survival or progression(recurrence) free survival data (Katagairi et al. Mod Pathol. 2012 Feb;25(2):282-8; Huang et al. Mod Pathol. 2014 Jul;27(7):983-90).

3) Patients with MMR mutations has a possibility to have a Lynch syndrome. Please consider to state if there is any patient who has a Lynch syndrome in this cohort (Ge et al. Diagn Pathol. 2021 Feb 4;16(1):12.)

  1. Discussion

1) In the discussion section, the authors stated that immunohistochemistry staining in a validation cohort showed concordant results. If the authors performed immunohistochemistry in this cohort, the data should be shown (Itamochi et al. Int J Clin Oncol. 2015 Oct;20(5):967-73).

2) There is similar result have published in the past (Shibuya et al. Genes Chromosomes Cancer. 2018 Feb;57(2):51-60.). This article showed APOBEC enrichment and MMR signatures using NGS. Please discuss differences, similarities and unique points of this article compared with Shibuya’s article.

Author Response

(The authors gave the same response as above.)

Round 2

Reviewer 1 Report

Thank You for the suitable changes. 

Author Response

We would like to thank the Reviewer for the critical points raised, and the enrichment of the discussion.

Reviewer 3 Report

The manuscript has been revised well and I agree that this manuscript described the significant contribution to distinguish genomic sub-classification of ovarian cancer using ARID1A and PIK3CA mutation profiles. However, there are minor concerns to be revised by the authors.

1) Considering the previously published manuscript by Shibuya and his colleagues (Shibuya et al. Genes Chromosomes Cancer. 2018 Feb;57(2):51-60), the authors should discuss the originality of this project. The authors described the difference of this study from Shibuya’s manuscript in the first revision, however, they described one of the differences is a use of a gene panel routinely applied in their clinical settings. If authors would like to confine this manuscript to the clinical usefulness of the gene panel, the authors should describe the relevance between the result and clinical background. On the other hand, if the authors would like to discuss the stratification of OCCC in molecular subgroups, the authors should describe the superiority of this manuscript compared with Shibuya’s report.

Author Response

Please find the corresponding reply attached.
